# Different Time Courses of Mono- and Bi-Liganded Bursts of Channel Openings of Adult nAChR Molecules Formed by the Reactions of Transmembrane Regions

**DOI:** 10.3390/cells13242079

**Published:** 2024-12-17

**Authors:** Dmitrij Ljaschenko, Martin Pauli, Achmed Mrestani, Josef Dudel, Manfred Heckmann

**Affiliations:** 1Institute of Physiology, Department of Neurophysiology, University of Würzburg, Röntgenring 9, 97070 Würzburg, Germany; dmitrij.ljaschenko@medizin.uni-leipzig.de (D.L.);; 2Rudolf Schönheimer Institute of Biochemistry, Division of General Biochemistry, Medical Faculty, Leipzig University, Johannisallee 30, 04103 Leipzig, Germany; 3Department of Neurology, Leipzig University Medical Center, 04103 Leipzig, Germany; 4Institute for Neuroscience, Technical University Munich, Biedersteinerstr. 29, 80802 München, Germany

**Keywords:** acetylcholine, burst, channel, endplate, kinetics, receptor, synapse

## Abstract

We recorded transmembrane currents through single nicotinic acetylcholine receptors (nAChRs) in cell-attached patches at high temporal resolutions from cultured and transiently transfected HEK 293 cells. Receptor activation was elicited by acetylcholine (ACh) or epibatidine (Ebd) at concentrations ranging from 0.01 to 100 µM, binding to one (R_αδ_ or R_αε_) or both extracellular ligand binding sites (R_αδ+αε_). Agonist binding to R_αδ_ resulted in very short openings with mean durations of (τ_o1_ < 5 µs), while the binding to R_αε_ produced short (τ_o2_ = 37 µs) and intermediate openings (τ_o3_ = 187 µs). Binding at both sites (R_αδ+αε_) generated long openings (τ_o4_ = 752 µs). All durations are noted in brackets since missed closures could shorten the results. Mono-liganded bursts were elicited at 0.01 µM ACh or Ebd, lasted less than a millisecond, displayed the typical current amplitude, and were interrupted by frequent microsecond-scale closures (µBs) that often did not reach the zero current. In contrast, bi-liganded bursts exhibited classical full amplitudes and long open states lasting up to several milliseconds, interspersed with rare µB closures of a similar duration to those observed in mono-liganded bursts.

## 1. Introduction

Among the most extensively studied receptors in humans are muscle nicotinic acetylcholine (ACh) receptors (nAChRs), with a diameter of approximately 10 nm [1]. There are over 10,000 nAChRs per µm^2^ in human endplates, resulting in millions of nAChRs in each endplate and trillions throughout our bodies. Adult nAChRs comprise five subunits, combining two α-, one β-, one δ-, and one ε-subunit. Structural alterations in the αδ- and αε-ligand binding sites, triggered by agonist activity, travel from the outer regions to the channel’s gate in the transmembrane domains (TMDs), spanning 6 nm, through a rapid succession of changes in conformation, leading to the activation of the nAChR [2].

Muscle nAChRs create both bi-liganded and short mono-liganded openings [3]. Moreover, the αδ- and αε-sites may play unequal roles in receptor gating. This disparity in contribution may also be reflected in the distinct opening patterns initiated by agonist binding at these sites in nAChRs. 

Precisely measuring the duration of brief openings and closures can be critical in studying the single-channel currents of nAChRs [4,5,6,7]. The duration of closed times in the mono-liganded bursts of nAChRs provides insights into the channel opening process. Although mono-liganded bursts have limited relevance compared to bi-liganded bursts of nAChRs in neuromuscular transmission [8,9,10], one may argue that they still are interesting. Despite the extensive research on nAChRs, our understanding of mono-liganded bursts of nAChRs still needs to be improved [11].

We conducted single-channel recordings of adult muscle nAChRs using ACh, the naturally occurring transmitter, and the frog poison epibatidine (Ebd), focusing on low noise and high temporal resolution. Regarding affinity, ACh and Ebd are almost on par with each other for nAChRs. However, Ebd’s larger head group volume produces less binding energy for gating than ACh [12]. A broad range of concentrations (ranging from 10 nM to 100 µM) of the two agonists was employed to differentiate between mono-liganded and bi-liganded bursts. We were surprised by the new information about the fine structure of mono-liganded bursts, which was visible with an increased recording resolution. Additionally, the effects of agonist binding at the αδ- and αε-sites of nAChRs were examined with the assistance of two natural compounds: snail-derived α-Conotoxin M1 (CTx) and snake venom Waglerin-1 (WTx); [13,14,15,16].

## 2. Materials and Methods

### 2.1. Receptor Expression in Cell Culture

HEK 293 cells were obtained from ATCC (CRL-1573), cultured on poly-L-lysine-coated coverslips at 37 °C, 5% CO_2_, in Dulbecco’s modified Eagle’s medium (Gibco, Thermofisher Scientific, Frankfurt, Germany). The medium was supplemented with 10% foetal calf serum (Gibco), 100 units/mL penicillin, and 100 µg/mL streptomycin (Gibco) [17]. Cells were transiently transfected to express murine adult nAChRs using the calcium phosphate co-precipitation method [18]. The murine muscle acetylcholine receptor subunit ε was encoded on a pcDNA3 plasmid [19], as well as subunits α, β, and δ on three pRc/CMV plasmids (ratio α, β, δ, ε: 2, 1, 1, 1). The GFP-carrying plasmid pmaxGFP (Amaxa, Lonza Group Ltd., Basel, Switzerland) was added to the transfection solution to mark transfected cells.

### 2.2. On-Cell Patch Clamping

Before recording, coverslips were transferred to recording chambers containing a physiological solution (in mM, 162 NaCl, 5.3 KCl, 2 CaCl_2_, 0.67 NaH_2_PO_4_, 15 HEPES, pH 7.4). Patch pipettes also contained either acetylcholine (ACh) at concentrations of 0.01, 0.1, 1, 10, and 100 µM or epibatidine (Ebd) at concentrations of 0.01, 0.1, and 1 µM. On-cell patch-clamp recordings were performed at room temperature (19.5–22.9 °C, mean: 21.7 ± 1.0 SD °C). All patches were hyperpolarized by 200 mV to improve the signal-to-noise ratio, while the resting potential of transfected HEK cells was about −30 mV, and the reversal potential of the nAChR current was −20 mV with our recording solutions.

In some experiments, αδ-sites were blocked. To this end, cells were incubated in 1 µM CTx-containing [15,20,21,22] culture medium for 10 min at 37 °C. After transferring the coverslip to the recording chamber, a CTx-free pipette was used for recording. Since a 50–150 mbar pressure was applied to each pipette before patch formation, access to CTx from the bath solution to the patch was prevented. All αδ-sites were blocked in most patches, and recording could start. This procedure was helpful since the concentration of CTx had to be high enough to block all αδ-sites rapidly, but the additional application of CTx to the patch would have blocked too many αε-sites. CTx dissociated from the αδ-site at a shallow rate during recording. Eventually, agonists bound to both binding sites, eliciting long bursts. Thus, as soon as the first long burst appeared, the evaluation of the recordings ended.

To block the αε-site of the receptor, WTx was applied [15,16] using the same routine as with CTx. The WTx measurements were performed after experimenting with different incubation periods, blockers, and agonist concentrations. WTx was only applied for five minutes to keep more αδ-sites unaffected. Additionally, the ACh concentration in the pipette was 10 µM to increase the number of events. 

### 2.3. Low-Noise Modifications

Axopatch 200B, coupled with a capacitive feedback head stage (Molecular Devices, San Jose, CA, USA), was connected to patch-clamp pipettes to record single-channel currents. Due to its exceptional electrical and surface properties, thick-walled quartz glass (2 mm outer diameter, 1 mm inner diameter) was used to manufacture pipettes [6,23]. With further modifications described in [6], a low noise level was achieved: 1.5 ± 0.1 (SD) pA root mean square (RMS) at 60 kHz (−3 dB) low-pass filtering, and 153 ± 30 (SD) fA RMS at 5 kHz (−3 dB) low-pass filtering. 

### 2.4. Data Storage and Filtering 

Currents were digitized using Signal 3 (Cambridge Electronic Design, Milton, UK) at 1 MHz with a Power 1401 mkII (CED) analogue–digital converter and stored as described in [22]. They were then processed and analysed further with the DC analysis software suite (David Colquhoun, University College London, https://github.com/DCPROGS accessed on 10 November 2024).

### 2.5. Dwell Time Distributions

Dwell time distributions were calculated using Ekdist (DC software, 1997 DOS program). The abscissae were logarithmically transformed to cover a wide range of dwell times. Exponential probability density functions (pdfs) were fitted to dwell time distributions, and since the ordinate was transformed by the square root, peaks in the resulting plot represent the time constants of the pdf [24,25,26]. Further details are given elsewhere [7,22]. The duration of t_crit_ was obtained by extrapolating the extent of the first shut-time component (τ_c1_) with a straight line fitted to the right shoulder of its exponential probability function and by determining the intersection of this line with the abscissa. This yielded 26 µs for ACh and 20 µs for Ebd.

### 2.6. Statistics 

If not stated otherwise, data are reported as mean ± SEM (standard error of the mean). The number of pdf components for the open period, single opening, and burst length distributions was obtained by increasing the number of components as long as the probability of erroneously accepting a non-existing component was below 1% [7,27,28].

## 3. Results

### 3.1. High-Resolution Recordings

Each experiment started by sealing a thick-walled quartz electrode on the membrane surface of cells expressing adult nAChRs to record the channel currents [6,7]. Currents through single adult muscle type nAChRs reached an amplitude of −15.0 ± 0.4 pA at −230 mV hyperpolarization (Figure 1). This signal amplitude allowed for low-pass filtering at 40 kHz for evaluation. At the same time, the signal-to-noise ratio remained at least 11.8, and the mean signal-to-noise ratio was 14.2 ± 1.6 in this study. Hence, the temporal resolution could be set at 5 µs in evaluating currents [26]. All recordings were scanned for the simultaneous openings of multiple channels during data evaluation. If simultaneous openings occurred, these data sections were not evaluated.

Figure 1 shows original currents with 0.1 µM ACh. Channel openings were downward. The uppermost trace shows 10 s of a recording. A grey horizontal line below the trace marks a 1-s-long region. The middle trace displays this 1-s-long region with a tenfold expanded time scale, and the below grey horizontal line marks a 100 ms long region. The bottom trace shows the 100 ms long region with a tenfold expanded time scale and displays a typical long bi-liganded burst of openings. In addition to this rather long receptor channel activity, many shorter events are discernible in the upper traces. We will analyse these shorter events in more detail below.

Figure 2 shows high-resolution currents filtered at 60 kHz. Channel openings are again downward. A typical long bi-liganded burst of openings recorded with 1 µM ACh is shown in Figure 2A. We spread the time course of the currents more than usual (note the scale bar of 100 µs in the lower right of the figure), and we were surprised by new insights. We present the duration of individual openings next to their beginning (in µs, red) to relate the currents to the evaluations in Figure 2E and Appendix A. Analogously, we marked the beginnings of closings with their durations using blue numbers. Note that the analysis of dwell time distribution was conducted on traces filtered at 40 kHz.

Figure 2A shows three sections of one 14.79 ms long burst of openings at 1 µM ACh elicited from a bi-liganded receptor: its beginning, middle section with two closings, and ending. Closings within bi-liganded bursts in response to ACh are relatively rare, and openings are long. In the example shown in Figure 2A, the burst has a total duration of 14.79 ms, and its evaluation revealed nine closures. Three are shown in Figure 2A; they are short, with durations of 5, 8, and 5 µs, and start steeply from the open channel current level. Figure 2B shows three segments of a second 7.04 ms long bi-liganded burst. Its evaluation revealed 11 openings and 10 closures. The start of the first opening is shown on the left. The section in the middle shows the longest gap in this burst, and on the right, the end of the last opening (and, thus, the end of this burst) is shown.

Figure 2C shows a very low ACh concentration of 0.01 µM, at which only mono-liganded receptor currents were observed (see also Figure 2E and Table 1). Compared to bi-liganded bursts, the open times of mono-liganded bursts are very short (Figure 2C). Openings and closings often do not reach the full amplitude due to limits in temporal resolution, and almost all openings are bordered by brief closings, which we refer to as micro blocks (µB), lasting usually only a few µs and which are, most of the time, not fully resolved. The first mono-liganded burst in Figure 2C contains four openings with durations of 72–181 µs (as determined at 40 kHz) and three relatively long µBs lasting, in this case, 9, 11, and 10 µs. The burst below has three µBs with 9, 5, and 7 µs durations. Also, the block of αδ-sites by CTx [13,20,21] in Figure 2D results in typical mono-liganded bursts. The left panel in Figure 2D shows a burst at high ACh concentration and a short burst with µBs with durations of 7 and 6 µs with CTx. The last burst on the right in Figure 2D is another even shorter example of a burst with two openings with durations of 31 and 21 µs and a single µB with a duration of 6 µs.

### 3.2. Open Periods and Shut Times with ACh and Ebd 

Figure 2E shows dwell time distributions of open periods from five sample experiments for a 10,000-fold range of ACh concentration, and mean values are given in Table 1. At 0.01 µM ACh, the open periods contain only two components (τ_o2_ and τ_o3_), peaking at 37 and 187 µs. The shut times shown in Appendix A present a prominent peak at 2.9 µs, reminding us of the breaks seen in the bi-liganded and mono-liganded bursts of Figure 1 and Figure 2. The second peak at about 1 s reveals that openings are rare at this low agonist concentration. At 0.1 µM ACh, two additional opening components (τ_o1_ and τ_o4_) appear at 1.9 and 787 µs. Examples of long openings (τ_o4_) are shown in the bi-liganded bursts of Figure 2A. At 1 µM ACh, short openings (τ_o2_) are less frequent, and intermediate (τ_o3_) open periods of about 200 µs are not resolved as a separate component. Some openings of this duration may have been obscured by the massive peak of long openings (τ_o4_) at 876 µs and, thus, may not have reached the level of statistical significance.

The picture changes further at higher ACh concentrations (10 and 100 µM). Again, only two components of open periods are resolved. The very short openings (τ_o1_) are diminished in frequency, and long openings (τ_o4_) are shortened from 876 µs at 1 µM ACh to 557 µs at 10 µM ACh and 213 µs at 100 µM ACh. At the latter concentration, the dominating 2 to 3 µs shut time is joined by a companion, a 7 µs component almost as prevalent as the briefest component (Appendix A).

Similar to Figure 2, we show examples of original recordings with the agonist epibatidine (Ebd) in Figure 3 [12,29]. Compared to Figure 2A, the bi-liganded burst in Figure 3A presents a much higher rate of closures. Mean values for open times with Ebd are given in Table 1. The mono-liganded bursts of openings in Figure 3B are not different from those in Figure 2B, except that they appear shorter. Like Figure 2, the open periods for Ebd are shown in Figure 3C, and the corresponding shut time distributions are shown in Appendix A. Ebd has a slightly higher affinity to the receptors than ACh. In contrast to ACh, it gives rise to a significant number of long openings (τ_o4_), which stem from bi-liganded nAChRs already at 0.01 µM (regarding affinity, see also Figure 2 in [12]. The 39 and 156 µs opening components with 0.01 µM Ebd are equivalents of τ_o2_ and τ_o3_ seen with ACh. At all concentrations of Ebd in Appendix A, a very short shut time is prominent. With 0.1 µM Ebd, three types of openings appear again, but now the long component (τ_o4_) is dominant (Figure 3C). Unlike that of the same ACh concentration, the very short open period (τ_o1_) is not resolved. At 1 µM Ebd, very short open periods are present (like with ACh), but again, like with ACh, τ_o3_ is unresolved. Interestingly, µBs occur less often at higher agonist concentrations and are much less frequent in bi-liganded than mono-liganded bursts. This appears to be the case for both Ebd and ACh.

### 3.3. Definition and Durations of Bursts

Openings of nAChRs appear either as single openings separated by long shut times or as bursts, i.e., openings separated by short closings [4]. A critical shut time (t_crit_) in quantitative evaluations distinguishes between bursts and single openings. All openings separated by closings shorter than or equal t_crit_ are considered openings within one burst. A closing longer than t_crit_ ends a burst. Openings flanked at both sides by closings longer than t_crit_ are considered single openings. The most prominent and best-studied activity in single-channel recordings of muscle nAChRs is the long bursts, generally thought to arise from nAChRs with two agonists bound. They consist of long openings (τ_o4_) separated by very short shut times. Long or bi-liganded bursts [5,30,31], contain agonist-independent brief closings. “Agonist-independent” refers to the duration of the closings; however, the length of the gaps is beyond our recording resolution, which precludes firm statements.

The duration of bi-liganded bursts decreases with increasing concentrations for both agonists (Figure 4, Table 2 and Appendix A). With 0.01 µM Ebd, the mean duration of mono-liganded bursts was significantly lower (277 ± 21 µs) than with 0.01 µM ACh (450 ± 62 µs, SD, n = 3 in each group, *p* = 0.03). In these recordings with Ebd, the mean τ_o2_ was 36 ± 4 µs and 37 ± 16 µs with ACh (*p* = 0.87). Furthermore, with Ebd, the mean τ_o3_ was 139 ± 15 µs and 188 ± 21 µs with ACh (*p* = 0.03). However, the mean duration of bi-liganded bursts was not different with 0.1 µM Ebd (5.9 ± 1.4 ms) from that with 0.1 µM ACh (11.8 ± 4.5 ms, SD, n = 3 in each group, *p* = 0.14).

### 3.4. Blocking the αδ- or αε-Site of nAChRs

Figure 5A,B show the evaluations from experiments with blocked αδ-sites of the receptors by incubating cells in 1 µM of the snail CTx (see also Figure 2D). This should reveal the channel activity generated by agonist binding at the αε-site of nAChRs. Both ACh and Ebd elicited short single openings (τ_o2_) but no intermediate openings (τ_o3_) and 100–130 µs mono-liganded bursts of openings (Figure 5A,B and Appendix A). Mono-liganded bursts with CTx are shorter than mono-liganded bursts without blockers and contain only short (τ_o2_) openings. We tested the agonist concentration dependence of the CTx block using higher ACh concentrations and obtained similar results with 0.1 to 10 µM ACh. With 100 µM ACh plus CTx, the mean duration of short openings was reduced. In addition, a significant fraction of very short openings was observed (Appendix A). The mean duration of mono-liganded bursts with 0.1 µM ACh plus CTx was significantly lower (102 ± 43 µs) than with 0.1µM ACh (547 ± 140 µs, SD, n = 3, *p* = 0.024). Likewise, with 0.1 µM Ebd plus CTx, the mean duration of mono-liganded bursts was again significantly lower (120 ± 14 µs) than with 0.1 µM Ebd (267 ± 25 µs, SD, n = 3, *p* = 0.003). However, the duration of mono-liganded bursts with 0.1 µM ACh plus CTx (102 ± 43 µs) was not different from that with 0.1 µM Ebd plus CTx (120 ± 14 µs, SD, n = 3, *p* = 0.55).

Waglerin-1 (WTx) is a snake venom that blocks neuromuscular transmission in mammals [14,15,16] preferentially at the αε-site of nAChRs. The data for a cell incubated with 1 µM WTx are shown in Figure 5C. The shut time plot indicates that openings are infrequent; on average, less than one occurred during one second. Furthermore, only very short openings occurred with a mean duration of 13 µs. Almost all openings were singles, but occasionally, bursts occurred, as illustrated in the right panel of Figure 5C.

## 4. Discussion

The nAChR molecules are activated by the binding of ACh or a similar agonist to one or both of its ligand binding sites (R_αδ_, R_αε_, and R_αδ+αε_, Figure 6). The simultaneous activation of αδ-and αε-sites elicits bi-liganded or αδ+αε-bursts of τ_o4_ openings [4]. The other options activate mono-liganded openings. When ACh is applied at a very low concentration of 0.01 µM, only τ_o2_ and τ_o3_ openings appear (Figure 2 and Table 1), while with 0.1 µM ACh, all four types of openings are observed. Thus, τ_o2_ and τ_o3_ are elicited at αε-sites and τ_o1_ at αδ-sites of receptors. These assumptions are supported by the mixed τ_o2_ and τ_o3_ bursts in Figure 2C, showing what appear to be instantaneous switches between τ_o2_ and τ_o3_ openings. Blocking the αδ-site of the receptor with CTx resulted in τ_o2_ openings, and blocking the αε-site of the receptor by WTx led to very short openings that may resemble τ_o1_ (Figure 5C). Some authors have seen spontaneous openings. We have not seen them, but excluding some accidental action by agonist molecules may be difficult. Spontaneous openings must have been rare, given the significant difference in activation with 0.01 µM ACh and Ebd.

### 4.1. Short Mono-Liganded Bursts and the Frequency of µBs

It is puzzling that so many of these brief µB closings exist, and why they exist is unclear. As a first notion, one might assume that the brief closings or µBs are simply noise without any physiological meaning. However, these short closings occur more frequently in mono-liganded than bi-liganded bursts (Figure 2 and Figure 3). In mono-liganded bursts, almost all openings are associated with µBs, which appear to be fundamentally linked to the openings. One might also assume that µBs are generated by permeating ions or deprotonations of amino acid side chains facing the channel lumen [32,33]. However, the finding that the µBs are more frequent in mono-liganded than in bi-liganded bursts argues against this notion. Ctx was expected to hinder receptor function by blocking the αδ-site, thus preventing bi-liganded receptor activation and the activation of the αδ-site. While bi-liganded bursts were, as desired, very efficiently stopped by CTx, the additional finding was unexpected that mono-liganded bursts were shortened by eliminating the intermediate open time component τ_o3_. The elimination of τ_o3_ by CTx (Figure 2 and Figure 5 and Appendix A) suggests that CTx bound at the αδ-site reduces the ability of agonists to elicit channel opening at the αε-site. However, it was found that even receptors of only αβδ subunits (presumed to be α_2_βδ_2_ receptors) generate bursts and recapitulate the kinetics of slow muscle currents in zebrafish [34].

Mono-liganded bursts are significantly shorter with Ebd than with ACh. Relating this to Ebd’s lower efficiency and larger head group volume is tempting. It will be interesting to test this for other agonists, such as carbamylcholine (CCh), a partial agonist with higher efficiency than Ebd, for which detailed measurements of efficiency in recordings of bi-liganded bursts at higher agonist concentrations are available [2,12]. Furthermore, studying the effects of additional agonists in the same efficiency class could allow for determining whether the differences are only associated with differences in efficacy. These measurements must be performed at sufficiently low agonist concentrations to obtain preferential mono-liganded bursts, like with 0.01 µM Ebd, or even better, exclusively mono-liganded bursts, like with 0.01 µM ACh. However, these measurements will not require a very high resolution since differences in burst duration, such as the one found by us (0.3 ms versus 0.5 ms for Ebd or ACh at 0.01 µM), should be readily quantifiable as agonist-dependent differences in open time with, e.g., 10 kHz recording resolution. It would also be interesting to extrapolate equilibrium dissociation constants (K_ds_) and calculate agonist efficiency for different agonists at each ligand binding site to clarify, e.g., whether a given agonist’s efficacy differs at the αδ- and the αε- ligand binding sites.

### 4.2. The Center of the Molecule

The center of the nAChR molecule is located within the synaptic muscle membrane and contains a channel that may pass the ionic current across the cell membrane. We were especially interested in the receptor activation of a single agonist, which generates mono-liganded bursts. If agonists bind simultaneously at αε and αδ, the classical and much longer bi-liganded or αδ+αε-bursts are elicited. 

The binding of ACh only at the αε-site opens the channel (O_αε_A in Figure 6). O_αε_A is terminated by the short closure µB. The activation should immediately return upon dissociating A to R_αε_ to trigger a single opening. After a microsecond block (µB), another opening O_αε_A needs to occur to generate a burst. This arrangement elicits different opening and closure programs from a specific receptor site. We have no evidence for its details, but it seems to be the simplest solution. However, more complex ones cannot be excluded. Schemes, such as Scheme 1 in [35], do not allow better descriptions of the mono-liganded bursts, and the rate constants suggested in Table 1 of Mukhtasimova et al. (2016) [35] predict the gaps to be longer in mono-liganded than in bi-liganded bursts.

In our recordings, the maximal amplitude open channel currents are strictly set to −15 pA. This description of the functions of the nAChR molecule’s center is also valid in principle for different agonists (Suberyldicholine, Epibatidine, and Carbachol) in adult and embryonic nAChRs [6,7,22].

### 4.3. High-Resolution Recordings of Bi- and Mono-Liganded Bursts of Openings

The sample records in Figure 2 and Figure 3 show that “bi- and mono-liganded” bursts differ in character. Bi-liganded bursts can last tens of milliseconds and are separated by very short and steep closures into few openings of, on average, about 1 ms duration (at low ACh concentrations, Figure 2). Mono-liganded bursts are much shorter (Figure 2B,C and Figure 4). They are separated into openings (τ_o2_ and τ_o3_) that may appear to start and end relatively slowly (compared to the bi-liganded ones) and often appear to not reach the full amplitude of −15 pA.

It has been argued that the short µBs are artifacts of the recording. However, the frequency of µB decreases with increasing concentrations of the agonist (see Figure 2 and Figure 3 for ACh and Ebd), which appears to be the case for other agonists. The regular and repetitive occurrence of µB in mono-liganded bursts suggests that µB is certainly not an artifact.

### 4.4. Desensitization

Our recordings were generated by single-nAChR molecules that react to ACh or Ebd, as depicted in Figure 6. While we used constant agonist concentrations, other patch-clamp experiments with fast agonist application on nAChR molecules resulted in sums of currents from hundreds of such molecules [10,36]. When the patch is exposed to a pulse of ACh at a concentration >1 µM, the elicited initial current pulse decays exponentially to a lower maintained level; it is “desensitized” (or open channel blocked). An ACh concentration of 100 µM will desensitize the current from the patch by a factor of about 100. In our present experiments on single receptor molecules, we expose the receptor molecule continuously to 100 µM ACh. But, we see no signs of desensitization; the recorded maximal current amplitude yielded a measure of the standard −15 pA (as shown by [37]). Desensitization reduces the number of activatable receptors, blocking many channels due to binding a high ACh concentration to the extracellular receptor milli blocks (mB, Figure 6). This open channel block at high ACh concentrations reduces the bursts and durations of single openings and longer activation desensitizes further openings.

### 4.5. Additional Shut Time

At our highest concentration of 100 µM ACh in this study, the extrapolated 3 µs shut time was joined by a relatively large 7 µs component (Appendix A). A concentration-dependent component should be initiated by the binding of ACh, but we have no information about its position in the scheme of Figure 6. We mention this unclear finding since it may relate to the open-channel block at high ACh concentrations [38]. In addition to the desensitization, some further blocking mechanisms might be operating. The extensive desensitization at high ACh concentrations shortens the bursts of channel openings that can elicit the depolarization and excitation of the muscle cell.

## 5. Conclusions

The juvenile type of nAChRs was also investigated using the same techniques as the present ones [22]. The results were similar to those for adult nAChRs; only the current amplitudes were lower than 15 pA, and the discussion was less detailed. 

The main function of the muscle nAChRs is to transmit the impulse of the motor nerve to the depolarization and excitation of the muscle fiber through long bursts of openings mediated through the bi-liganded receptor. The functions of the mono-liganded receptors need to be clarified. At very low ambient serum ACh concentrations, the relatively small currents generated by short mono-liganded openings might help establish and stabilize the nerve–muscle synapse [39,40]. The channel opening mechanism and the possibility of channel blocks allow for optimal adaptations to functional needs.

## Figures and Tables

**Figure 1 cells-13-02079-f001:**
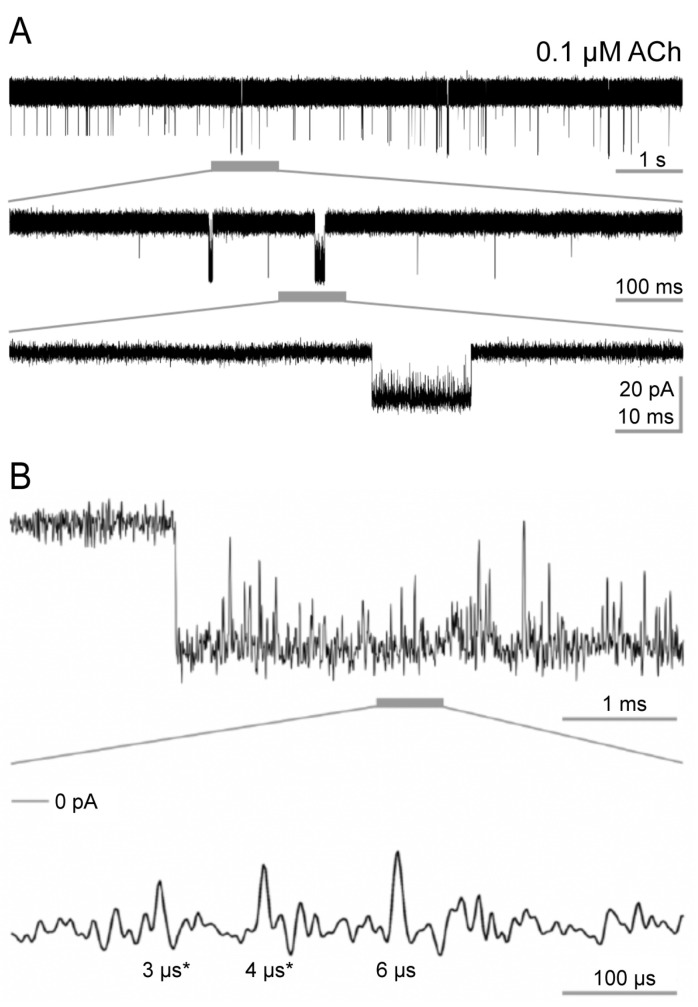
Examples of acetylcholine (ACh)-elicited channel openings. (**A**) Single channel currents were recorded at 22 °C with thick-walled quartz pipettes in the on-cell patch-clamp configuration from transiently transfected HEK 293 cells. The traces were acquired with 0.1 μM ACh and low-pass filtered at 40 kHz. Sections of interest are underlined with horizontal grey bars and displayed below with a tenfold expanded time scale. The lower panel shows a typical long burst. In addition to such rather long receptor channel activity, many much shorter events are discernible in the upper traces. (**B**) Examples of channel closings during a long burst elicited by 0.1 μM ACh. The tenfold expanded section displayed in the lower panel contains three gaps. Below each gap, its duration is given. The first two are 3 µs and 4 µs long, marked by an asterisk (*), and did not pass our 5 µs threshold for temporal resolution (see first paragraph of Section 3.1). The third gap is 6 µs long. Above the lower trace, the zero current level is indicated on the left.

**Figure 2 cells-13-02079-f002:**
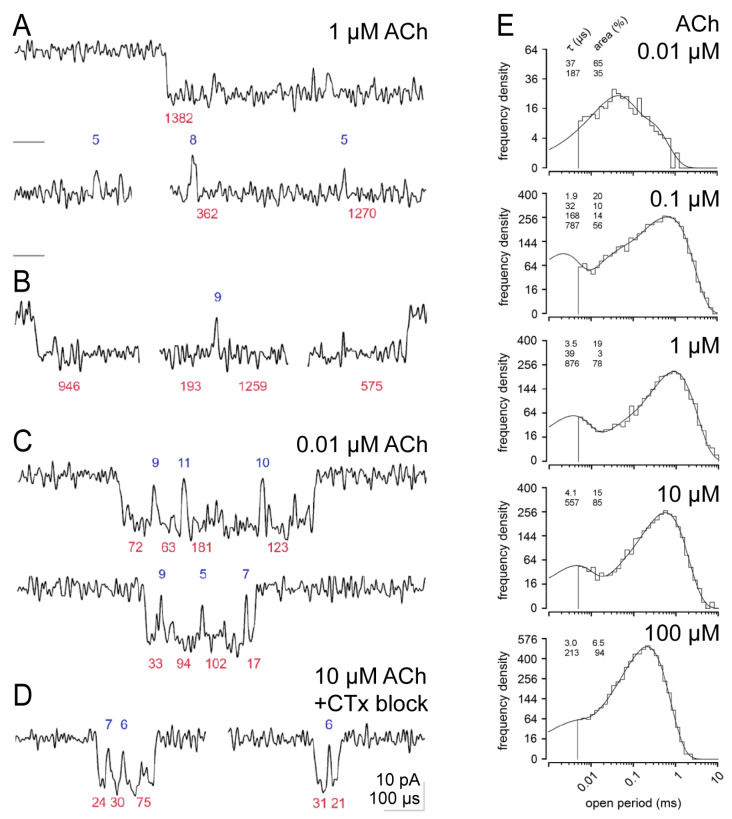
Original records of bursts with ACh at 60 kHz temporal resolution and open period distributions. Below and above the current traces, open times (red) and shut times (blue, both in µs) characterize openings and closings in the adjacent current trace. The respective numerical values were gained in evaluations of open and closed times, like in Figure 3 at 40 kHz. All current traces have the same calibration of amplitude and time. (**A**) Three sections of a 14.79 ms long burst of openings elicited by a bi-liganded receptor (with 1 µM ACh), showing its initial 1382 µs long opening, which is terminated by a 5 µs long gap which is shown on the left in the middle panel. The horizontal straight line on the left in this panel indicates the closed level at 0 pA. The third section towards the end of this burst shows an 8 µs long gap before a 362 µs long opening, another 5 µs long gap, and then a 1270 µs long opening in the right panel, which terminates this long burst in the lowest panel. (**B**) Three sections of a 7.04 ms burst of openings from the same recording. The segment on the left shows the start of the first opening, the middle segment shows the longest gap in this burst, and the third segment on the right shows the end of the final opening in this burst. (**C**) Examples of short mono-liganded bursts elicited by 0.01 µM ACh. (**D**) Mono-liganded bursts with 10 µM ACh while the R_αδ_ receptor site is blocked by α-Conotoxin-M1. (**E**) Probability density functions fitted to open period distributions from recordings with 0.01 to 100 µM ACh. ACh concentration increases from top to bottom, as indicated on the right. Ordinates give the number of events per bin and the axis of abscissae event durations on a log scale. Distributions of open periods were fitted by 2–4 component probability density functions (see Section 2.5). The parameters τ and areas of each fit are given in the upper left of each plot. Very short and long openings appear at 0.01 µM ACh and higher concentrations. Corresponding distributions of shut times are shown in Appendix A.

**Figure 3 cells-13-02079-f003:**
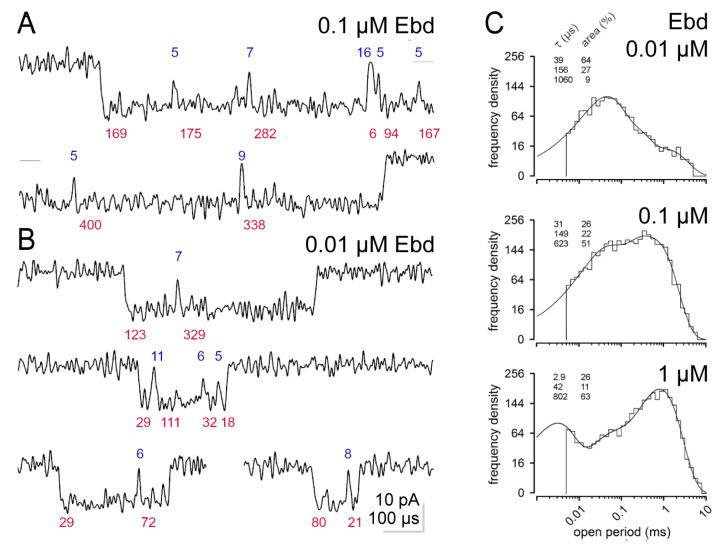
Original records of bursts with Ebd at 60 kHz and open period distributions. Below and above the traces, open times (red) and shut times (blue, both in µs) are given for the adjacent current like in Figure 2. All traces have the same amplitude and time calibrations. (**A**) Two sections of a 4.01 ms long burst of openings elicited by a bi-liganded receptor (0.1 µM Ebd), showing its beginning and end. A short horizontal line at the right side of the upper panel and the left side of the lower panel indicates the closed level at 0 pA. (**B**) Four examples of short mono-liganded bursts elicited by 0.01 µM Ebd. (**C**) Probability density functions fitted to open period distributions from recordings at 0.01 to 1 µM epibatidine (Ebd). Same arrangement as in Figure 2. Long openings already appear with 0.01 µM Ebd, but very short openings require higher Ebd concentrations.

**Figure 4 cells-13-02079-f004:**
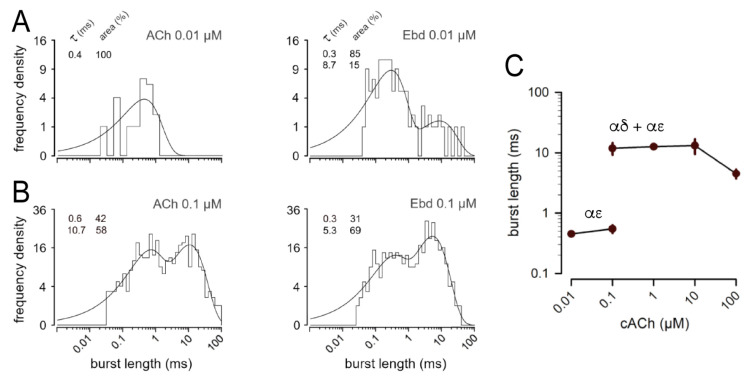
Burst length distributions with ACh and Ebd. For clarity, single openings were removed from burst length distributions. Time constants of the fits (τ; µs) and their proportion (area; %) are shown on the upper left of each plot. (**A**) Durations of bursts (t_crit_ = 26 µs) with 10 nM ACh (**left**) and with 10 nM Ebd (t_crit_ = 20 µs, **right**). With ACh, the distribution was fitted with one component: 0.4 ms, 100%. With Ebd, the distribution was fitted with 2 components: 0.3 ms, 85%; 8.7 ms, 15%. (**B**) Durations of bursts (t_crit_ = 26 µs) with 0.1 µM ACh (**left**) and with 0.1 µM Ebd (t_crit_ = 20 µs, **right**). With ACh, the distribution was fitted with 2 components: 0.6 ms, 42%; 10.7 ms, 58%. With Ebd, the distribution was fitted with 2 components: 0.3 ms, 31%; 5.3 ms, 69%. (**C**) Concentration dependence of the duration of mono-liganded (αε) and bi-liganded (αδ+αε) bursts. Mean duration ± SEM in recordings with ACh (3 recordings for each concentration as shown in Table 2).

**Figure 5 cells-13-02079-f005:**
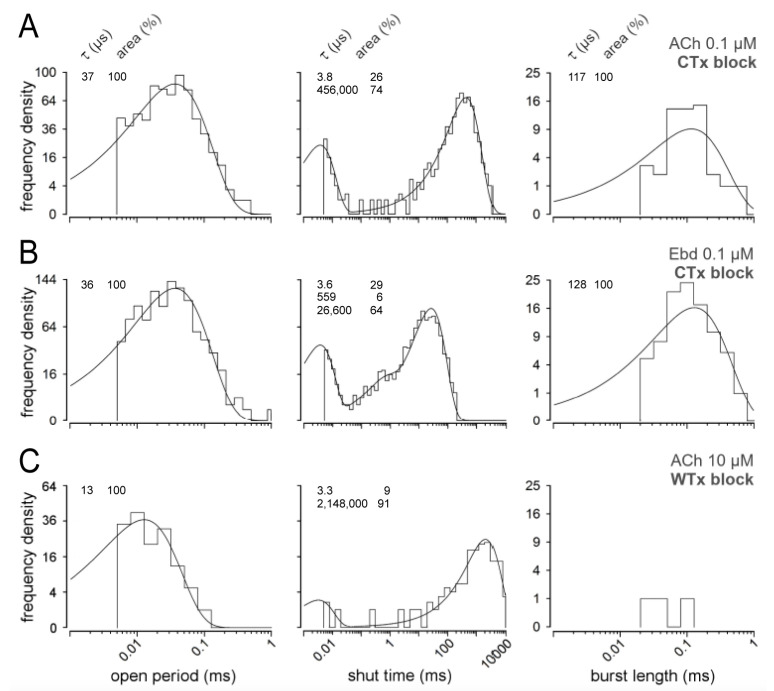
Partial block of nAChRs. Dwell time distributions from ACh or Ebd recordings after incubation in CTx or WTx, specific blockers of the αδ-site and αε-site, respectively. Open period, shut time, and burst length distributions with 0.1 µM ACh (**A**) or Ebd (**B**) after incubation with 1 µM CTx. Open period distributions were fitted with a one-component pdf. Shut time distributions were fitted with two (ACh) or three (Ebd) components. For clarity, single openings were removed from burst length distributions. Burst length distributions were fitted with one component. Time constants of the fits (τ; µs) and their proportion (area; %) are shown on the upper left of each plot. (**C**) Open period, shut time, and burst length distributions with 10 µM ACh after incubation with 1 µM WTx.

**Figure 6 cells-13-02079-f006:**
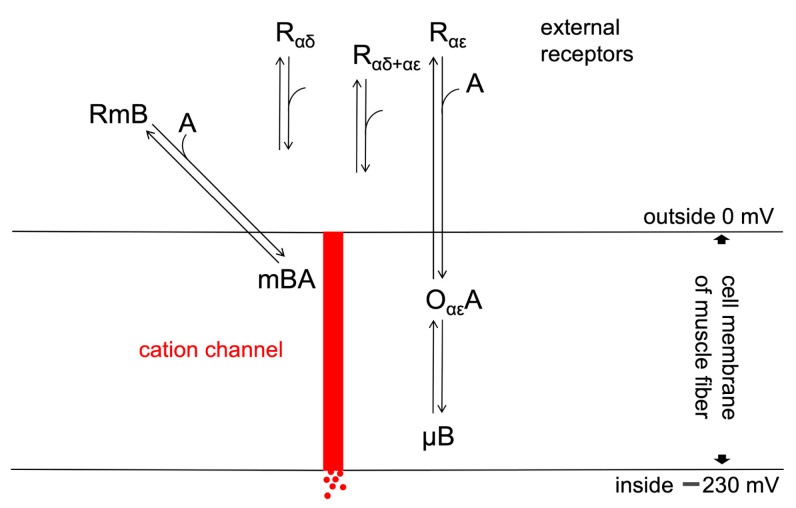
Scheme of the AChR molecule exposed to the agonist ACh. The receptor (R) binds the agonist (A) at its ligand binding site (R_αε_). This elicits a conformational change that reaches the receptor’s central channel, initiating openings (O_αε_A) that can be followed by microsecond-long channel blocks (µB). An additional blocking mechanism named mB is also elicited by ACh binding. This occurs at higher concentrations, blocks channel openings completely, lasts several milliseconds, and is responsible for the open channel block or desensitization mechanisms.

**Table 1 cells-13-02079-t001:** Mean open periods for ACh and Ebd. Open periods of adult mouse muscle type nAChRs at the indicated ACh and Ebd concentrations. Mean values of time constants with standard deviations from fits of distributions from three recordings are given in microseconds. Mean component areas are shown in %.

	Very Short Openings (τ_o1_)		Short Openings (τ_o2_)		Intermediate Openings (τ_o3_)		Long Openings (τ_o4_)	
ACh (µM)	τ_o1_ (µs)	Area (%)	τ_o2_ (µs)	Area (%)	τ_o3_ (µs)	Area (%)	τ_o4_ (µs)	Area (%)
0.01			37 ± 16	51 ± 12	188 ± 21	49 ± 12		
0.1	2.3 ± 0.4	18 ± 10	30 ± 9	10 ± 1	176 ± 11	14 ± 0	891 ± 446	63 ± 19
1	2.8 ± 0.7	16 ± 10	58 ± 26	4.5 ± 2.1			819 ± 413	81 ± 14
10	3.8 ± 0.3	13 ± 2					546 ± 29	87 ± 2
100	2.9 ± 0.4	9 ± 4					209 ± 9	92 ± 4
Ebd (µM)								
0.01			36 ± 4.2	59 ± 10	139 ± 15	27 ± 2	1240 ± 171	14 ± 9
0.1			45 ± 15	27 ± 6			521 ± 188	59 ± 17
1	3.4 ± 0.8	28 ± 3	45 ± 20	9 ± 2			616 ± 191	72 ± 18

**Table 2 cells-13-02079-t002:** Mean burst durations with short t_crit_ for ACh and Ebd. Burst durations of adult mouse muscle type nAChRs with ACh or Ebd. Mean values of time constants with standard deviations from fits of distributions from 3 recordings are given in milliseconds. Mean component areas are shown in %. In the recordings with ACh, t_crit_ was 26 µs and 20 µs with Ebd. For clarity, single openings were removed from burst length distributions.

	Short (αε) Bursts		Long (αδ+αε) Bursts	
ACh (µM)	τ_b1_ (ms)	Area (%)	τ_b2_ (ms)	Area (%)
0.01	0.45 ± 0.06	100 ± 0		
0.1	0.55 ± 0.14	33 ± 19	11.8 ± 4.5	67 ± 19
1			12.6 ± 0.3	100 ± 0
10			13.1 ± 6.3	100 ± 0
100			4.5 ± 1.3	100 ± 0
Ebd (µM)				
0.01	0.29 ± 0.03	66 ± 20	8.50 ± 0.53	34 ± 20
0.1	0.27 ± 0.03	39 ± 12	5.93 ± 1.36	61 ± 12
1			4.23 ± 0.45	100 ± 0

## Data Availability

The authors will make the raw data supporting this article’s conclusions available upon request.

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
