# Peer review of "Different Time Courses of Mono- and Bi-Liganded Bursts of Channel Openings of Adult nAChR Molecules Formed by the Reactions of Transmembrane Regions"

_cells, 2024, doi:10.3390/cells13242079_

Round 1

Reviewer 1 Report

Comments and Suggestions for Authors

This is another insightful contribution from Heckmann lab, where they employ high-resolution patch-clamp techniques with high-cutoff frequency filtering to achieve a temporal resolution of 5 µs. This approach enabled the observation of micro-closures (μBs) and distinct gating dynamics in muscle-type nicotinic receptors during mono- and bi-liganded states. The study represents high-quality work with significant findings on sub-conductance states and differences in agonist action, providing valuable insights into receptor allostery. It is particularly appreciated that authors do not prematurely associate their observations, like the brief closures to any conformational state and letting the data speak for itself.

The choice of ACh and Ebt as agonists is particularly appropriate as they have similar affinities but different efficacy and hence different efficiencies, allowing meaning comparison with similar drug concentrations. It would be interesting to see if lower frequency of µBs in Ebt correlated with its lower agonist efficiency. Examining whether CCh, also a partial but higher efficiency agonist, has more frequent µB like ACh might provide an answer. This is also true with lower burst times for lower eta class agonists, is it consistent with other agonists in the same eta class and not associated with efficacy alone. If so, it would be an interesting find.

Given the measurements were done at different agonist concentrations for each agonist binding site, Can Kds be extrapolated and have you thought of calculating agonist efficiency at each site? It will be interesting to see if different eta class agonists have different etas for the two sites. Atleast discussing this would be of interest.

Author Response

Reviewer 1

Comments and Suggestions for Authors

The manuscript of Ljaschenko et al. entitled:” Different time courses of mono- and bi-liganded bursts of channel openings of adult nAChRs molecules as formed by the reactions of the transmembrane regions” characterizes the gating behavior of the adult-type mouse muscle nicotinic acetylcholine receptor (nAChR) expressed in the HEK293 cells.  The authors focused on mono-liganded bursts, which occur when only one of the two activation sites is occupied. To investigate this phenomenon, they employed a sophisticated experimental approach using two antagonists, each specific to one activation site, to compete with activator binding. While the physiological relevance of mono-liganded bursts remains unclear, the authors suggested they might stabilize the nerve-muscle synapse at rest. This intriguing possibility underscores the importance of studying this phenomenon further. Although the manuscript advances our understanding of nAChR gating, it would benefit from being presented in a more reader-friendly manner. In particular, the Results section is overly detailed, which can make it challenging for readers to maintain focus. Reorganizing the figures could greatly enhance the manuscript's accessibility. In conclusion, the manuscript is promising, and I strongly encourage the authors to revise it to maximize its impact.

Thank you for reviewing our manuscript and for your constructive suggestions. As detailed below, we followed your suggestions and reorganized the manuscript.

Specific concerns and recommendations

  1. When CTx was used to block the δ site, even mono-liganded bursts were shortened. This was unexpected, but the authors did not provide any commentary on this phenomenon. Additionally, it would be interesting to activate the channel with a higher concentration of acetylcholine (ACh) in the presence of CTx to determine whether the duration of mono-liganded bursts would increase. The data suggest that the δ site modulates the dominant ε site. This conclusion is supported by the observation that when WTx was used to occupy the ε site, channel activity decreased, and no bursts were detected. This indicates that the δ site alone is insufficient to induce bursts. However, a previous study (www.jgp.org/cgi/doi/10.1085/jgp.201110649) demonstrated that when the ε subunit was replaced by a δ subunit, creating a channel consisting of two δ subunits, bursts were still observed—and even prolonged. This phenomenon should be discussed by the authors.

We added a comment to the Results section on 9, line 324, and in the discussion on pages 10-11, line 382-86, regarding the shortened burst with Ctx. We also added a table with data from recordings with CTx and ACh at various concentrations (new Table S2). Furthermore, we included the suggested reference and referred to its finding in the revised Discussion on page 11, line 386.

  1. How did the authors confirm that only a single channel was recorded in each patch? This should be clearly stated, as the presence of multiple channels with low activity could negatively impact the closed-time analysis

We added a sentence to the Results section on pages 3-4, lines 140-42.

  1. Lines 424-429: The authors speculated that “…….the ion channel contains two parallel sections addressed to open and close either from the δ- or the ε-site of the receptors. The flow of ions through these half-channels should be less effective than the parallel flow through both halves elicited from simultaneous agonist binding at the δ- and ε-site of nAChRs….”.

I do not agree, even if it is a speculation, because if ions were flowing through only one half-channel pore, the current would be halved. However, this was not observed.

This paragraph was deleted. See page 12 line 436.

  1. I recommend to enlarge Figure 1 to clearly show the short openings and closings during the burst duration.

Figure 1 was enlarged with a second panel (B) to clearly show the transitions during the burst (See page 4 Figure 1).

  1. Figures 2 and 3 could be combined. The closed-time distributions from Figure 3 could be moved to the Supplementary Materials. The representative traces in Figure 2 should be smaller and displayed on a larger time scale to highlight the slow and long bursts in relation to ACh concentration. Table S1 for ACh should be moved from the Supplementary Materials into the main text.

Figures 2 and 3 were combined as suggested (see new Figure 2 on page 5), and the closed time distribution moved to Figure S1. Furthermore, the original Table S1 was moved to the main text (see Table 1 on page 6).

Similarly, Figures 4 and 5 should be reorganized for clarity.

Done as suggested (see new Figure 3 on page 7 and Figure S2 in the Supplement).

  1. Figures 6 and 7 could be combined, and Table S2 should be moved from the Supplementary Materials into the main text.

Done as suggested (see new Figure 4 on page 8 and Table 2 on page 9).

  1. In Figure 8, the closed-time distributions could be moved to the Supplementary Materials, and representative traces should be added for each experimental conditions. Average open and closed times in the presence of CTx and WTx should also be presented in tables.

A Table with average open times (see new Table S2) was added, but we decided not to rearrange this Figure (now Figure 5 on page 10) since representative traces are already shown in Figures 2 and 3.

Reviewer 2 Report

Comments and Suggestions for Authors

Needs substantial improvement

The manuscript by Ljaschenko et al. presents an interesting single-channel patch-clamp study that adds valuable insight into the function of muscle-type nicotinic acetylcholine receptors (nAChRs). However, some revisions are needed before it can be considered for publication. In particular, the manuscript currently includes some inaccuracies regarding the identification of the orthosteric binding sites. Specifically, the authors appear to attribute these binding sites to the epsilon (ε) and delta (δ) subunits of the adult muscle-type nAChR. However, the binding sites for acetylcholine (ACh) are located at the α-ε and α-δ interfaces, rather than on the individual ε and δ subunits. Clarification and correction of this point will be important.

Additionally, the nomenclature used to refer to the agonist-binding sites is somewhat confusing. The terms Rε, Rδ, and Rεδ” are used as if they represent distinct nAChR isoforms with individual ε or δ subunits (the two former) and a hybrid species (the latter), but this could lead to misinterpretation. Then, because the peculiar notation should be attributed to the electrophysiological fingerprint of the agonist-elicited responses when a single site or the two orthosteric sites are activated, respectively. This needs to be spelled out in clear form in the manuscript. These issues are especially evident in the Discussion section, where the authors state: The nAChR molecules are activated by the binding of ACh or a similar agonist to one or both of its extracellular receptors (Rδ, Rε, and Rδε). It would be more accurate to refer to these as the canonical orthosteric binding sites, rather than as separate receptors. Reference to existing literature, such as reference 22, would help to resolve this ambiguity.

Abstract

The abstract could benefit from some refinement in syntax and clarity. In particular, the authors should explicitly mention that the study was conducted in a transient expression mammalian cell system, which is an important detail for readers to understand the experimental context.

Introduction

Building on the earlier points, the introduction should introduce the correct nomenclature for the ligand recognition sites of the adult muscle-type nAChRs. Currently, the nomenclature is inconsistent with accepted terminology. For example, works such as Goswami et al. (2023) Nature Communications 14:3169; Li et al. (2024) Nature 633:1174-1180. provide a useful reference for proper nomenclature and should be consulted to ensure accuracy. In addition, the authors should clarify their notation of the agonist-activated receptors, substituting their current acronyms Rε, Rδ, and Rεδ by a new set of abbreviations, unambiguously explaining that they are referring to monoliganded and bi-liganded receptors and not receptor isoforms.

Results

The quality of the patch-clamp single-channel recordings is very high, and the temporal resolution is excellent. The authors describe using 1 µM snail conotoxin (CTx) to block the αδ site and generate channel openings specifically at the αε site, as well as using waglerin-1 (WTx) to target the αε site and produce channel activity via the αδ site. While these experiments are informative, it would be helpful to know whether the authors have explored other concentrations of the toxins. As these toxins are not highly selective, varying concentrations could lead to some overlap in their targeting of the agonist-binding sites, which might affect the interpretation of the data.

Figure 9

In the figure legend for Figure 9, the term AChR molecule should be used rather than AChRs molecule, as the latter is grammatically incorrect.

Comments on the Quality of English Language

Needs improvement

Author Response

Reviewer 2

Comments and Suggestions for Authors

The manuscript by Ljaschenko et al. presents an interesting single-channel patch-clamp study that adds valuable insight into the function of muscle-type nicotinic acetylcholine receptors (nAChRs). However, some revisions are needed before it can be considered for publication. In particular, the manuscript currently includes some inaccuracies regarding the identification of the orthosteric binding sites. Specifically, the authors appear to attribute these binding sites to the epsilon (ε) and delta (δ) subunits of the adult muscle-type nAChR. However, the binding sites for acetylcholine (ACh) are located at the α-ε and α-δ interfaces, rather than on the individual ε and δsubunits. Clarification and correction of this point will be important.

Additionally, the nomenclature used to refer to the agonist-binding sites is somewhat confusing. The terms Rε, Rδ, and Rεδ are used as if they represent distinct nAChR isoforms with individual ε or δ subunits (the two former) and a hybrid species (the latter), but this could lead to misinterpretation. Then, because the peculiar notation should be attributed to the electrophysiological fingerprint of the agonist-elicited responses when a single site or the two orthosteric sites are activated, respectively. This needs to be spelled out in clear form in the manuscript. These issues are especially evident in the Discussion section, where the authors state: The nAChR molecules are activated by the binding of ACh or a similar agonist to one or both of its extracellular receptors (Rδ, Rε, and Rδε). It would be more accurate to refer to these as the canonical orthosteric binding sites, rather than as separate receptors. Reference to existing literature, such as reference 22, would help to resolve this ambiguity.

Thank you for reviewing our manuscript and for your constructive comments. Following your suggestions, we reorganized the designation of the ligand binding sites in the entire manuscript, the new Figure 4 (on page 8), Figure 6 (on page 11), and the new Table 2 (on page 9). To make clear that ligands bind at the interfaces of the subunits, we refer now to αδ- αε- ligand binding sites. Furthermore, the above-quoted first sentence from our discussion was modified (page 9, line 347).

Abstract The abstract could benefit from some refinement in syntax and clarity. In particular, the authors should explicitly mention that the study was conducted in a transient expression mammalian cell system, which is an important detail for readers to understand the experimental context.

The abstract was refined, and we mention now (lines 15-16) the transient expression and the cell system.

Introduction Building on the earlier points, the introduction should introduce the correct nomenclature for the ligand recognition sites of the adult muscle-type nAChRs. Currently, the nomenclature is inconsistent with accepted terminology. For example, works such as Goswami et al. (2023) Nature Communications 14:3169; Li et al. (2024) Nature 633:1174-1180. provide a useful reference for proper nomenclature and should be consulted to ensure accuracy. In addition, the authors should clarify their notation of the agonist-activated receptors, substituting their current acronyms Rε, Rδ, and Rεδ by a new set of abbreviations, unambiguously explaining that they are referring to monoliganded and bi-liganded receptors and not receptor isoforms.

As mentioned above, we clarified the nomenclature in the entire manuscript. We introduced a new set of abbreviations to explain unambiguously that we are referring to mono-liganded and bi-liganded receptors and not receptor isoforms.

Results The quality of the patch-clamp single-channel recordings is very high, and the temporal resolution is excellent. The authors describe using 1 µM snail conotoxin (CTx) to block the αδ site and generate channel openings specifically at the αε site, as well as using waglerin-1 (WTx) to target the αε site and produce channel activity via the αδ site. While these experiments are informative, it would be helpful to know whether the authors have explored other concentrations of the toxins. As these toxins are not highly selective, varying concentrations could lead to some overlap in their targeting of the agonist-binding sites, which might affect the interpretation of the data.

We did not explore other concentrations of the toxins, and we see no evidence for overlap in their targeting in our data, but we agree that this might occur.

Figure 9 In the figure legend for Figure 9, the term AChR molecule should be used rather than AChRs molecule, as the latter is grammatically incorrect.

Done, page 11, line 392

Reviewer 3 Report

Comments and Suggestions for Authors

The manuscript of Ljaschenko et al. entitled:” Different time courses of mono- and bi-liganded bursts of channel openings of adult nAChRs molecules as formed by the reactions of the transmembrane regions” characterizes the gating behavior of the adult-type mouse muscle nicotinic acetylcholine receptor (nAChR) expressed in the HEK293 cells.  The authors focused on mono-liganded bursts, which occur when only one of the two activation sites is occupied. To investigate this phenomenon, they employed a sophisticated experimental approach using two antagonists, each specific to one activation site, to compete with activator binding. While the physiological relevance of mono-liganded bursts remains unclear, the authors suggested they might stabilize the nerve-muscle synapse at rest. This intriguing possibility underscores the importance of studying this phenomenon further. Although the manuscript advances our understanding of nAChR gating, it would benefit from being presented in a more reader-friendly manner. In particular, the Results section is overly detailed, which can make it challenging for readers to maintain focus. Reorganizing the figures could greatly enhance the manuscript's accessibility. In conclusion, the manuscript is promising, and I strongly encourage the authors to revise it to maximize its impact.

Specific concerns and recommendations

1. When CTx was used to block the δ site, even mono-liganded bursts were shortened. This was unexpected, but the authors did not provide any commentary on this phenomenon. Additionally, it would be interesting to activate the channel with a higher concentration of acetylcholine (ACh) in the presence of CTx to determine whether the duration of mono-liganded bursts would increase. The data suggest that the δ site modulates the dominant ε site. This conclusion is supported by the observation that when WTx was used to occupy the ε site, channel activity decreased, and no bursts were detected. This indicates that the δ site alone is insufficient to induce bursts. However, a previous study (www.jgp.org/cgi/doi/10.1085/jgp.201110649) demonstrated that when the ε subunit was replaced by a δ subunit, creating a channel consisting of two δ subunits, bursts were still observed—and even prolonged. This phenomenon should be discussed by the authors.

2. How did the authors confirm that only a single channel was recorded in each patch? This should be clearly stated, as the presence of multiple channels with low activity could negatively impact the closed-time analysis

3. Lines 424-429: The authors speculated that “…….the ion channel contains two parallel sections addressed to open and close either from the δ- or the ε-site of the receptors. The flow of ions through these half-channels should be less effective than the parallel flow through both halves elicited from simultaneous agonist binding at the δ- and ε-site of nAChRs….”.

I do not agree, even if it is a speculation, because if ions were flowing through only one half-channel pore, the current would be halved. However, this was not observed.

4. I recommend to enlarge Figure 1 to clearly show the short openings and closings during the burst duration.

5. Figures 2 and 3 could be combined. The closed-time distributions from Figure 3 could be moved to the Supplementary Materials. The representative traces in Figure 2 should be smaller and displayed on a larger time scale to highlight the slow and long bursts in relation to ACh concentration. Table S1 for ACh should be moved from the Supplementary Materials into the main text. Similarly, Figures 4 and 5 should be reorganized for clarity.

6. Figures 6 and 7 could be combined, and Table S2 should be moved from the Supplementary Materials into the main text.

7. In Figure 8, the closed-time distributions could be moved to the Supplementary Materials, and representative traces should be added for each experimental conditions. Average open and closed times in the presence of CTx and WTx should also be presented in tables.

Author Response

Reviewer 3

This is another insightful contribution from Heckmann lab, where they employ high-resolution patch-clamp techniques with high-cutoff frequency filtering to achieve a temporal resolution of 5 μs. This approach enabled the observation of micro-closures (μBs) and distinct gating dynamics in muscle-type nicotinic receptors during mono- and bi-liganded states. The study represents high-quality work with significant findings on sub-conductance states and differences in agonist action, providing valuable insights into receptor allostery. It is particularly appreciated that authors do not prematurely associate their observations, like the brief closures to any conformational state and letting the data speak for itself.

Thank you for reviewing our manuscript and for your constructive feedback.

The choice of ACh and Ebt as agonists is particularly appropriate as they have similar affinities but different efficacy and hence different efficiencies, allowing meaning comparison with similar drug concentrations. It would be interesting to see if lower frequency of μBs in Ebt correlated with its lower agonist efficiency. Examining whether CCh, also a partial but higher efficiency agonist, has more frequent μB like ACh might provide an answer. This is also true with lower burst times for lower eta class agonists, is it consistent with other agonists in the same eta class and not associated with efficacy alone. If so, it would be an interesting find.

Following your suggestion regarding CCh and other agonists, we discussed this idea on page 11, lines 401-2 and 403-5.

Given the measurements were done at different agonist concentrations for each agonist binding site, Can Kds be extrapolated and have you thought of calculating agonist efficiency at each site? It will be interesting to see if different eta class agonists have different etas for the two sites. Atleast discussing this would be of interest.

This is a suggestion that we incorporated into our discussion on page 11, lines 411-15.